# Emerging carbapenem-resistant *Klebsiella pneumoniae* in a tertiary care hospital in Lima, Peru

Fiorella Krapp,[1,2,3] Diego Cuicapuza,[2,4] Guillermo Salvatierra,[4] Jean P. Buteau,[2] Catherine Amaro,[5] Lizeth Astocondor,[1] Noemí Hinostroza,[1] Jan Jacobs,[3,6] Coralith García,[1,2,7] Pablo Tsukayama[1,4,8]

**ABSTRACT** The emergence of carbapenem-resistant *Klebsiella pneumoniae* (CRKP) poses a significant public health threat, particularly in low- and middle-income countries (LMICs) with limited surveillance and treatment options. This study examines the genetic diversity, resistance patterns, and transmission dynamics of 66 CRKP isolates recovered over 5 years (2015–2019) after the first case of CRKP was identified at a tertiary care hospital in Lima, Peru. Our findings reveal a shift from $bla_{KPC-2}$ to $bla_{NDM-1}$ as the dominant carbapenemase gene after 2017. Lineage ST45 was the most prevalent and persisted for multiple years, followed by high-risk clones ST11 and ST147. The $bla_{NDM-1}$ gene was carried almost exclusively by a Tn125-like transposon, similar to the one reported in previous studies from two Peruvian hospitals. Long-read sequencing revealed nearly identical $bla_{NDM}$-carrying plasmids across the four assessed lineages. A comparative analysis of 1,023 South American CRKP genomes confirmed a unique pattern in Peru, where $bla_{NDM-1}$ (81.4%) outpaced $bla_{KPC-2}$, which remained dominant (59.4%) elsewhere. In addition, emerging clones ST45 and ST348 found in Peru were rarely found elsewhere in South America, suggesting potential regional adaptation. In conclusion, our study provides a comprehensive picture of the intra-hospital dynamics of these emerging pathogens and provides a framework for studying their genomic diversity in the understudied South American region.

**IMPORTANCE** This study provides novel insights into the transmission and genetic diversity of carbapenem-resistant *Klebsiella pneumoniae*, a bacteria responsible for severe infections, with limited treatment options. By examining isolates recovered over 5 years at a major hospital in Lima, Peru, we demonstrated a shift from one type of resistance gene, $bla_{KPC}$, to another, $bla_{NDM}$, which is more challenging to treat. Our findings reveal that specific bacterial lineages carrying the $bla_{NDM}$ gene in a specific plasmid are emerging in Peru, including well-known high-risk strains and others rarely found elsewhere in South America. This pattern highlights an urgent need for targeted surveillance and infection control as these strains pose a significant challenge to healthcare systems. Our study provides crucial data on *Klebsiella pneumoniae* in Peru, contributing to broader efforts to monitor and control antibiotic-resistant infections in South America and globally.

**KEYWORDS** *Klebsiella pneumoniae*, carbapenem resistance, multidrug resistance, whole-genome sequencing, KPC-2, NDM-1

T he global spread of carbapenem-resistant *Klebsiella pneumoniae* (CRKP) infections poses a critical public health threat due to their limited therapeutic options and elevated mortality risk (1, 2). This threat is greater for low- and middle-income countries (LMICs), where antimicrobial resistance (AMR) surveillance, infection prevention and control strategies, and access to new antibiotics are limited (2, 3). In Latin America, the

**Peer Reviewer** Zhen Shen, Shanghai Jiao Tong University School of Medicine, Shanghai, China

Address correspondence to Fiorella Krapp, fiorella.krapp@upch.pe.

Fiorella Krapp and Diego Cuicapuza contributed equally to this article. The order of names reflects the leadership role taken by Fiorella Krapp in initiating and coordinating the project.

The authors declare no conflict of interest.

See the funding table on p. 12.

prevalence of CRKP isolates increased from less than 1% in 2007 to 16% in 2016 (4) and >20% in 2021 (2, 5).

Although Peru reports one of the highest rates of resistance to third-generation cephalosporins among *K. pneumoniae* in the region (2, 6), the emergence of CRKP in this country lagged behind the broader regional trend. The first CRKP case was reported in 2013 (7), and the frequency of CRKP remained below 10% through 2016 (4). However, between 2017 and 2019, a steep increase in the rate of carbapenem resistance was reported nationally (6, 8). Understanding the drivers of the emergence and dissemination of CRKP clones is essential for informing effective infection control strategies to limit the spread of this pathogen and prevent the emergence of other bacterial threats.

Whole-genome sequencing (WGS) has enabled high-resolution analyses of CRKP transmission and evolution at local (9), national (10–12), and regional scales (13, 14), by identifying high-risk clones, emerging mechanisms of resistance, and potential pathways for transmission. In Peru, WGS was used in 2016 to investigate one of the first documented outbreaks of NDM-producing *K. pneumoniae* in a tertiary care hospital in Lima (15). This study reported a multiclonal emergence of NDM-producing *K. pneumoniae*, encompassing four sequence types (ST348, ST11, ST147, and ST405), all carrying a novel $bla_{NDM-1}$-containing plasmid. Around the same time, other hospitals in Lima reported the emergence of CRKP, with ST348 identified as a prominent lineage, distinct from recognized global high-risk lineages (16).

To further investigate the factors driving CRKP emergence, we conducted a study at a tertiary care hospital in Lima during the first 5 years (2015–2019) following the first CRKP report. Using WGS, we assessed the diversity, resistance, and virulence profile of these emerging strains. We then compared these data with a systematically curated collection of publicly available CRKP sequences from Peru and South America to contextualize this emergence within the broader regional landscape.

## MATERIALS AND METHODS

### Bacterial isolates

We conducted a retrospective study at Hospital Cayetano Heredia (HCH), a 320-bed tertiary care teaching hospital that serves a population of over 2 million in northeast Lima. Since the first recovery of carbapenem-resistant Enterobacterales (CRE), the hospital laboratory stored all CRE isolates in skim milk at −20°C. Carbapenem resistance was defined as resistance to at least one carbapenem determined during routine testing (VITEK 2 Compact or disc diffusion) using the Clinical and Laboratory Standards Institute (CLSI) breakpoints valid at the time of testing. After Institutional Review Board approval by Universidad Peruana Cayetano Heredia and HCH, we reviewed the Microbiology Laboratory notebooks to identify all CRKP isolates grown as part of routine patient care from January 2015 to December 2019. We then retrieved three sets of CRKP isolates: i) all consecutive CRKP clinical isolates (*n* = 59) recovered during the first 3 calendar years (2015–2017); ii) three CRKP isolates recovered in a cross-sectional survey of hospital surfaces at the medical intensive care unit (ICU) conducted by the Epidemiology Office during November 2017 (Supplemental material); iii) CRKP clinical and rectal isolates recovered during snapshot periods during the next 2 calendar years (March 2018 (*n* = 12) and November 2019 (*n* = 8)) (Fig. S1). All CRKP isolates were stored in glycerol at −80°C and then processed at the Instituto de Medicina Tropical Alexander von Humboldt laboratories.

### Clinical data collection

For all retrieved isolates, we collected data on age, sex, culture sampling date, culture source, and hospital ward. For those belonging to the 2015–2017 period, we additionally conducted a retrospective chart review to collect data on the history of prior hospitalization, admission to the ICU, hospitalization outcome, and empiric antibiotic treatment

(within 2 days before and 2 days after sampling) to provide a proxy of "suspicion of infection" if antibiotic treatment was started vs "suspicion of colonization" if no antibiotic treatment was started by the treating physician. Two criteria were used as potential epidemiological links: i) a temporal distance of less than 3 months between recovery of isolates and ii) sharing a same ward of origin, defined as the hospital ward where the patient was hospitalized when the culture sample was obtained.

## Phenotypic testing

Only the first isolate per patient was included in the study. Stored isolates were reactivated on tryptic soy agar (TSA) and re-grown on MacConkey agar to confirm their purity, followed by manual biochemical identification and antimicrobial susceptibility testing. Susceptibility to meropenem, imipenem, ertapenem, and a panel of seven other antimicrobials was tested using the disk diffusion method (17). Susceptibility to colistin was tested by broth dilution and agar-spot method following the INEI-ANLIS protocol (18). CLSI 2021 breakpoints were used to determine resistance (19). Carbapenem resistance was confirmed if resistance to at least one of the three carbapenems tested was found. We used *Escherichia coli* ATCC 25922, *Pseudomonas aeruginosa* ATCC 27853, and *Proteus mirabilis* ATCC 12453 as control isolates.

## WGS and bioinformatic analysis

Confirmed carbapenem-resistant colonies were grown in tryptic soy broth (TSB) for 8 hours with continuous shaking, and DNA was extracted using the GeneJET Genomic DNA Purification Kit (Thermo Fisher). Illumina libraries were prepared from 1 ng of gDNA using the Nextera XT protocol and sequenced on a MiSeq instrument using v2 500-cycle sequencing kits. Raw reads were assessed with FastQC v0.12 (20), processed with Trimmomatic v0.39 (21), and assembled with Spades v3.15.5 (22). Assemblies were run through Kleborate v2.1 (23) for species identification, sequence typing, resistome profiling, and virulence gene detection. Genomes with novel alleles were submitted to BIGSdb (https://pubmlst.org/software/bigsdb) for analysis and sequence type (ST) assignment. AMRFinderPlus (24) was used to identify carbapenemase genes with less than 100% identity match in Kleborate. For expanded investigation of colistin resistance mechanisms, the coding sequences of the *pmrA*, *pmrB*, *phoP*, *phoQ*, *mgrB*, and *crrB* genes were manually extracted and aligned against *K. pneumoniae* NTUH-K2044 to look for mutations. To compare gene syntenies in carbapenemase-carrying plasmids, contigs with $bla_{KPC-2}$ and $bla_{NDM-1}$ genes were clustered at 90% sequence identity using CD-HIT-EST (-c 0.90 n 8) (25), genes were annotated, and their function was predicted using blastx (26). Panaroo v.1.3.3 (27) was used for pan-genome analysis and core genome-based phylogenies. Gubbins v3.3.0 (28) was used for the removal of recombinant sites and the construction of phylogenies, with references to *K. pneumoniae* NTUH-2044 (NC_012731.1), *K. quasipneumoniae* subsp. *quasipneumoniae* ATCC 700603 (CP014696.2), and *K. quasipneumoniae* subsp. *similipneumoniae* MGH44 (NZ_KI535595.1). Maximum-likelihood phylogenies were inferred with RAxML v1.2.0 (29) using the GTRGAMMA model with 100 iterations. Phylogenetic trees were visualized and annotated with iTOL v6.7.4 (30). We used snp-dists v0.8.2 (31) to calculate pairwise single-nucleotide polymorphism (SNP) distances from core genome alignments and used cutoffs of 10 or 15 SNPs (32, 33) to identify putative transmission clusters. Visualization of SNP distribution and clusters was performed using GraphSNP v1.0 (34).

To investigate the NDM-1-encoding plasmids, eight $bla_{NDM-1}$-carrying isolates from different ST groups and different years were selected for long-read Nanopore sequencing to fully resolve their chromosomes and plasmids. Libraries were prepared from 5 ng of DNA using the Rapid PCR Barcoding Kit and sequenced on an R10.4.1 flow cell with an Oxford Nanopore Technologies MinION Mk1B instrument. Raw Nanopore reads were trimmed with Porechop v0.2.4 (github.com/rrwick/porechop). *De novo* assemblies were generated using Dragonflye v1.2.1 (github.com/rpetit3/dragonflye), polished with Illumina reads using Polypolish v0.6.0 (github.com/rrwick/polypolish), and

reoriented with Dnaapler v0.8.1 (github.com/gbouras13/dnaapler). Plasmid sequences were aligned, and sequence similarity was assessed at 70%, 90%, and 100% identity thresholds using blastn (26). MOB-suite v3.9.1 (github.com/phac-nml/mob-suite) was employed to type the plasmids and predict their conjugative potential.

## Comparison to South American carbapenemase-producing *K. pneumoniae* sequences

A systematic search was conducted to identify all publicly available genome sequences of carbapenemase-producing *K. pneumoniae* in South America. Two search strategies were applied: i) we accessed NCBI Pathogen Detection Database (35) and filtered all clinical *K. pneumoniae* sequences uploaded between January 2000 and November 2022, with "location" specified as one of the 14 South American countries or territories and carrying at least one carbapenemase gene. ii) We systematically searched in PubMed, LILACS, and SciELO for articles published between January 2000 and December 2022, that included genomic sequencing data of CRKP clinical isolates recovered in any of the 14 South American countries or territories (search terms and steps are provided in the Supplementary materials). Identified articles were reviewed to determine if their sequences were publicly available. We retrieved GenBank or ENA accession numbers from these publications and downloaded available fasta assemblies or raw fastq files when available. *De novo* assemblies were generated from raw reads with the PathogenWatch web server (https://pathogen.watch/) (36), and all assemblies were then processed with Kleborate v2.1 (23). We filtered out sequences that were duplicated; had no identifiable carbapenemase gene; were isolated from an environmental source; or were not classified as *K. pneumoniae sensu stricto*. We also applied a quality control of the sequences and excluded sequences that were assembled into ≥300 contigs; had an N50 ≤50 000 bp; had ≥1,000 ambiguous bases; or had assembly errors. The resulting set of genomes was then compared against the *K. pneumoniae* sequences generated in this study that met the same quality criteria. We ran Panaroo v1.3.3 (27) to generate core-genome alignments and estimated maximum-likelihood phylogenies in RAxML (GTR model, 100 iterations). The resulting Newick tree was visualized and annotated in iTol v6.7.4 (30).

## RESULTS

### CRKP epidemic increase at a tertiary care hospital in Lima

The first clinical CRKP isolate was recovered from a blood culture in July 2015. Four additional cases were reported in 2015, seven in 2016, and 47 in 2017. The epidemic curve did not decrease during the 2 subsequent years involving the study, with 59 and 45 clinical isolates reported in 2018 and 2019, respectively. Cases were distributed across multiple hospital wards, including the emergency department and outpatient services, from the beginning of the emergence (Fig. 1). In November 2017, as part of the increased containment efforts, rectal screening of CRKP was introduced for ICU patients and contacts of positive cases in the hospitalization wards, resulting in 45, 362, and 189 rectal isolates of CRKP recovered during the years 2017, 2018, and 2019, respectively (Fig. 1). This measure was implemented intermittently due to limited hospital resources.

### Resistance phenotypes

We successfully regrew 43 out of 59 (73%) clinical CRKP isolates recovered from 2015 to 2017. Additionally, all 23 isolates from the two other sets (see *Methods*) were regrown, resulting in 66 unique CRKP isolates for analysis (Fig. S1). Co-resistance to non-β-lactam antibiotics was found in more than 65% of the isolates, except for colistin and amikacin, which retained *in vitro* activity in more than 90% of the isolates (Table 1). Notably, four isolates (6.1%) were resistant to colistin, with one exhibiting a minimum inhibitory concentration (MIC) of 4 µg/mL and three showing MICs greater than 4 µg/mL.

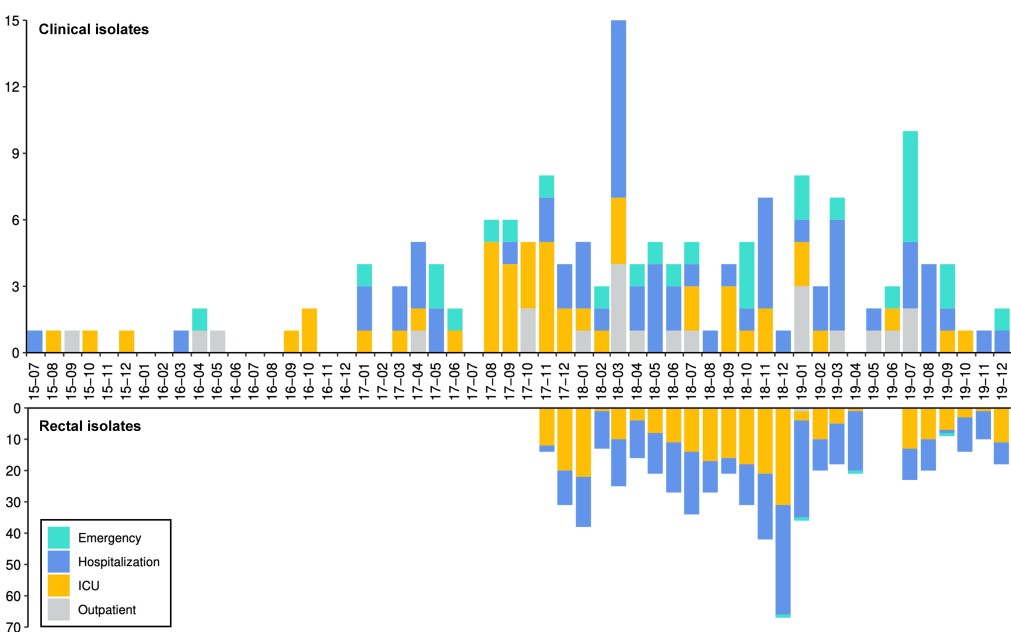

**FIG 1** Number of clinical isolates (top) and rectal isolates (bottom) of carbapenem-resistant *Klebsiella pneumoniae* (CRKP) recovered at the hospital's clinical microbiology laboratory from 2015 to 2019.

## Clinical characteristics

Retrieved isolates were recovered from urine ($n = 22$), blood ($n = 11$), respiratory secretions ($n = 9$), rectal swabs ($n = 14$), and other body secretions ($n = 7$) (Table 2). The median age of the patients was 53 years (interquartile range 35–62.5), with 50 patients (79.4%) hospitalized, including 28 who required ICU admission. The overall in-hospital all-cause case fatality was 32.3%. Antibiotic treatment was not initiated in 36.1% of patients, likely due to suspicion of colonization rather than infection (Table 2).

## Genomic diversity

Four species and subspecies of the genus *Klebsiella* were identified: *K. pneumoniae sensu stricto* ($n = 50$), *K. quasipneumoniae* subsp. *quasipneumoniae* ($n = 11$), *K. quasipneumoniae* subsp. *similipneumoniae* ($n = 4$), and *K. michiganensis* ($n = 1$) (Fig. 2). *K. quasipneumoniae* subsp. *quasipneumoniae* was the only species in 2015 and then remained the most common in 2016 (six out of seven isolates). In 2017, however, it was displaced by *K. pneumoniae sensu stricto*, which became the dominant species (28 out of 33 isolates). The 50 *K. pneumoniae sensu stricto* genomes were classified into 16 STs. The most prevalent was ST45 ($n = 20$), identified across 3 consecutive years (2017–2019) and in all the hospital surface isolates ($n = 3$). ST11 ($n = 6$) was the second most prevalent, mainly recovered in 2018–2019, followed by ST147 ($n = 5$) and ST15 ($n = 3$), found exclusively in 2017 (Fig. 3). Two genomes were classified as a novel ST (ST7693), both carrying $bla_{NDM-1}$.

## Potential transmission events

SNP distances between isolates ranged from 1 to 18,164, but it was less than 40 SNPs within each of the four most frequent ST groups, with ST147 and ST45 presenting the lowest within-group SNP distances (Fig. S2). Using a 10-SNP cutoff to infer potential transmission events, we identified six putative genetic clusters of 2–17 isolates. A 15-SNP

**TABLE 1** Antimicrobial co-resistance among 66 carbapenem-resistant *Klebsiella pneumoniae* isolates recovered at a tertiary care hospital in Lima, Peru

| Antibiotic | N° of resistant isolates (*N* = 66) |
| --- | --- |
| | n (%) |
| Beta-lactams | |
| Ertapenem | 63 (95.5) |
| Meropenem | 60 (90.9) |
| Imipenem | 48 (72.7) |
| Piperacillin/tazobactam | 62 (93.9) |
| Aztreonam | 60 (90.9) |
| Non-beta-lactams | |
| Ciprofloxacin | 56 (84.8) |
| Trimethoprim/sulfamethoxazole | 43 (65.2) |
| Gentamicin | 53 (80.3) |
| Amikacin | 3 (4.5) |
| Colistin | 4 (6.1) |
| Co-resistance to last-resort antibiotics | |
| Colistin +amikacin | 0 (0.0) |

threshold did not increase the number of clusters, but expanded the largest cluster to 20 isolates (Fig. S3). Among those clustered sequences, 52% (17/33) had a genetically nearest neighbor with two epidemiological links (recovered within 3 months and from the same hospital ward). An additional 27% (9/33) of clustered sequences had only one of the two epidemiological links. The largest genetic cluster included 17 ST45 CRKP isolates retrieved from the emergency department, ICU, and hospitalization wards between September 2017 and November 2019. Of these, 13 had a genetically nearest neighbor with two epidemiological links. Eight isolates in this cluster had an ICU isolate (KP129) as their closest genetic neighbor (Fig. S4).

## Antimicrobial resistance genes

The most common carbapenem resistance gene was $bla_{NDM-1}$, found in 46/66 isolates (69.7%), followed by $bla_{KPC-2}$ in 14 (21.2%) and $bla_{IMP-74}$ in five (7.6%). Only one isolate (KP136) did not have a detectable carbapenemase gene, and no dual-carbapenemase isolates were identified. The distribution of carbapenemase genes varied by *Klebsiella* species and over time, with $bla_{KPC-2}$ being the sole carbapenemase from 2015 to 2016, while $bla_{NDM-1}$ became predominant from 2017 to 2019 (Fig. 3). In addition to the carbapenemase genes, we identified other 16 β-lactamase genes, including five extended-spectrum β-lactamases (ESBLs) and 11 broad-spectrum β-lactamase genes, as well as 35 non-β-lactamase resistance genes (Fig. 2). Co-carriage of ESBL genes was found in 58 (87.9%) isolates. The most prevalent ESBL genes were $bla_{CTX-M-15}$ (38/66) and $bla_{TEM-26}$ (29/66). At least one non-β-lactam AMR gene was found in 63 (95.5%) isolates. The most common ones were the *aac(6')-Ib* aminoglycoside resistance gene found in 55 (83.3%) sequences, and the *sul1* sulfonamide resistance gene found in 51 (77.3%) (Fig. 3). The *mcr* gene family was absent in all the four colistin-resistant isolates. However, three of these isolates had resistance-associated mutations in the *pmrB* gene (T157P, A246T, and R256G), while the fourth isolate exhibited a V17E mutation in the mgrB gene and lacked the crrB gene entirely. No resistance-associated mutations were identified in the *pmrA*, *phoP*, or *phoQ* genes.

## Virulence determinants

No known virulence genes were identified in *K. quasipneumoniae* or *K. michiganensis* isolates. In contrast, 34 (68%) of the *K. pneumoniae sensu stricto* carried virulence genes. The yersiniabactin biosynthesis loci (*ybt*) were present in 29 (58%) sequences, with two variants: *ybt10* (*n* = 23) and *ybt9* (*n* = 6). The aerobactin biosynthesis locus *iuc*5 was also identified in five (10%) isolates. None of the sequences carried both virulence

**TABLE 2** Demographic, clinical, and microbiological characteristics of 63 patients with carbapenem-resistant *Klebsiella pneumoniae* infections at a tertiary care hospital in Lima, Peru

| Patient characteristics | Frequency n (%) |
|---|---|
| Demographic characteristics (*n* = 63) | |
| Sex, male | 49 (77.8) |
| Age (years)[a] | 53 (35–62.5) |
| Hospital location | |
| Ambulatory | 7 (11.1) |
| Emergency Department | 6 (9.5) |
| Hospitalization | 22 (34.9) |
| Intensive care unit | 28 (44.4) |
| Source of isolation | |
| Urine | 22 (34.9) |
| Stool | 14 (22.2) |
| Blood | 11 (17.5) |
| Respiratory | 9 (14.3) |
| Wound secretion | 3 (4.8) |
| Abdominal | 2 (3.2) |
| Drain | 1 (1.6) |
| Pleural | 1 (1.6) |
| Clinical characteristics (*n* = 42) | |
| Prior hospitalization | 14 (33.3) |
| Hospitalized | 36 (85.7) |
| Antibiotic treatment was not started | 13 (36.1) |
| Intensive care unit admission | 23 (63.9) |
| Died during hospitalization (*n* = 31) | 10 (32.3) |

[a]Median (interquartile range).

factors simultaneously. We did not detect salmochelin, colibactin, or *rmpACD* genes. All five sequences carrying *iuc5* belonged to ST147, while all the sequences carrying *ybt9* belonged to ST11. The remaining *ybt10*-carrying sequences were distributed across ST45 and ST15 (Fig. 3).

## Genetic context of carbapenemase genes

Genetic clustering of Illumina-based assemblies at 90% sequence found three representative sequences of $bla_{KPC}$-bearing contigs. The most frequent (*n* = 11) corresponded to the Tn4401b transposon and was identified in all the *K. quasipneumoniae* subsp. *quasipneumoniae* sequences (Fig. S5). Two clusters of $bla_{NDM}$-bearing contigs containing a Tn125-like backbone transposon were also identified. The largest cluster (*n* = 45) comprised sequences from multiple *Klebsiella* species and ST groups and presented a Tn125 transposon harboring the $bla_{NDM}$ gene and two other transposons, Tn6925 and Tn6308. The Tn6308 transposon also carried five additional resistance genes, including one encoding an aminoglycoside modifying enzyme and the *qacED1* gene associated with resistance to quaternary ammonium compounds (Fig. S5). This sequence also clustered with $bla_{NDM}$-bearing contigs reported in previous studies from two other hospitals in Lima (15, 37).

## NDM-1 encoding plasmids

The eight fully resolved $bla_{NDM}$-carrying IncFIB-IncHI1B plasmids found in four distinct *K. pneumoniae* lineages (ST11, ST15, ST45, and ST147) were approximately 372 kb in length, nearly identical to each other, and contained multiple additional resistance genes (Fig. 4). These plasmids showed high sequence similarity to a previously reported conjugative

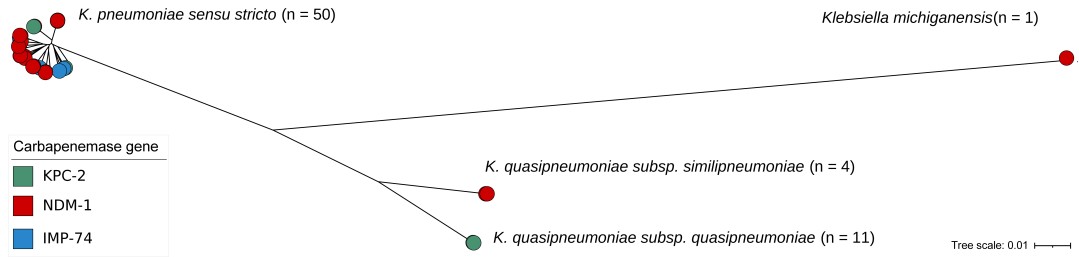

**FIG 2** Core-genome-based phylogeny of 66 *Klebsiella* spp. isolates recovered in this study. Node colors indicate the carbapenem resistance mechanism.

$bla_{NDM-1}$-carrying plasmid identified in a *K. pneumoniae* ST147 isolate from another hospital in Lima in 2017 (37).

## Comparison with CRKP sequences from South America

We analyzed 1,023 publicly available CRKP sequences from eight countries in South America (Fig. S6 to S8). Colombia contributed the most sequences (*n* = 538, 52.6%), followed by Brazil (*n* = 385, 37.6%), Argentina (*n* = 48, 4.7%), and Peru (*n* = 22, 2.2%). No CRKP sequences were available for Bolivia, Guyana, Paraguay, or Suriname. Most sequences (90.8%) were from isolates collected between 2013 and 2020. This data set comprised 120 ST groups, although 80% of the sequences were distributed in 16 ST groups. Among Peruvian sequences (*n* = 70, including 48 from this study), ST45 was the most prevalent lineage, followed by ST147, ST11, and ST348. In contrast, ST11 and ST258 were the dominant ST groups in other South American countries, and ST45 and ST147 were present in less than 2% of these sequences (Fig. 5; Fig. S9). Similarly, the most prevalent carbapenemase gene in Peruvian sequences was $bla_{NDM-1}$ (81.4%), whereas $bla_{KPC-2}$ was the most frequent carbapenemase gene (59.4%) in the other countries, with $bla_{NDM-1}$ present in only 9.3% of analyzed sequences (Fig. 5; Table S1).

## DISCUSSION

Analysis of consecutive CRKP isolates recovered during the first 5 years of CRKP emergence at a tertiary care hospital in Lima (2015–2019) revealed a multispecies and multiclonal emergence and dissemination. We observed that multiple KPC-2-producing *Klebsiella* species were initially introduced into the hospital and were later replaced by *K. pneumoniae sensu stricto* clones (i.e., ST45, ST11, and ST147) carrying the $bla_{NDM-1}$ gene. This replacement of KPC-producing isolates by NDM-producing isolates has been reported previously in other countries (38, 39). Because metallo-beta-lactamases (such as NDM) confer resistance to multiple antibiotics effective against CRKP (40), a global shift toward NDM dominance could further limit treatment options, underscoring the need for a coordinated global response to address this emerging threat. Locally, the predominance of NDM-producing CRKP limits the utility of the recently introduced antibiotic ceftazidime/avibactam as monotherapy, highlighting an urgent need to introduce newer antibiotics effective against metallo-beta-lactamases. Additionally, these findings underscore the need for more precise phenotypic or molecular methods for carbapenemase characterization in Peruvian hospitals to support patient care and epidemiological surveillance.

Our SNP analysis shows that nearly identical isolates were recovered from multiple patients and hospital wards, many of them with confirmed epidemiological links. This finding suggests transmission of CRKP within the hospital, although we cannot determine whether transmission occurred from patient to patient or involved intermediate hosts or surfaces. Nevertheless, the lack of electronic medical records limited our capacity to retrospectively assess other epidemiological links such as the overlap of hospital wards, surgical rooms, or procedures between the source patients during their entire hospitalization. Notably, the largest putative cluster, consisting of ST45

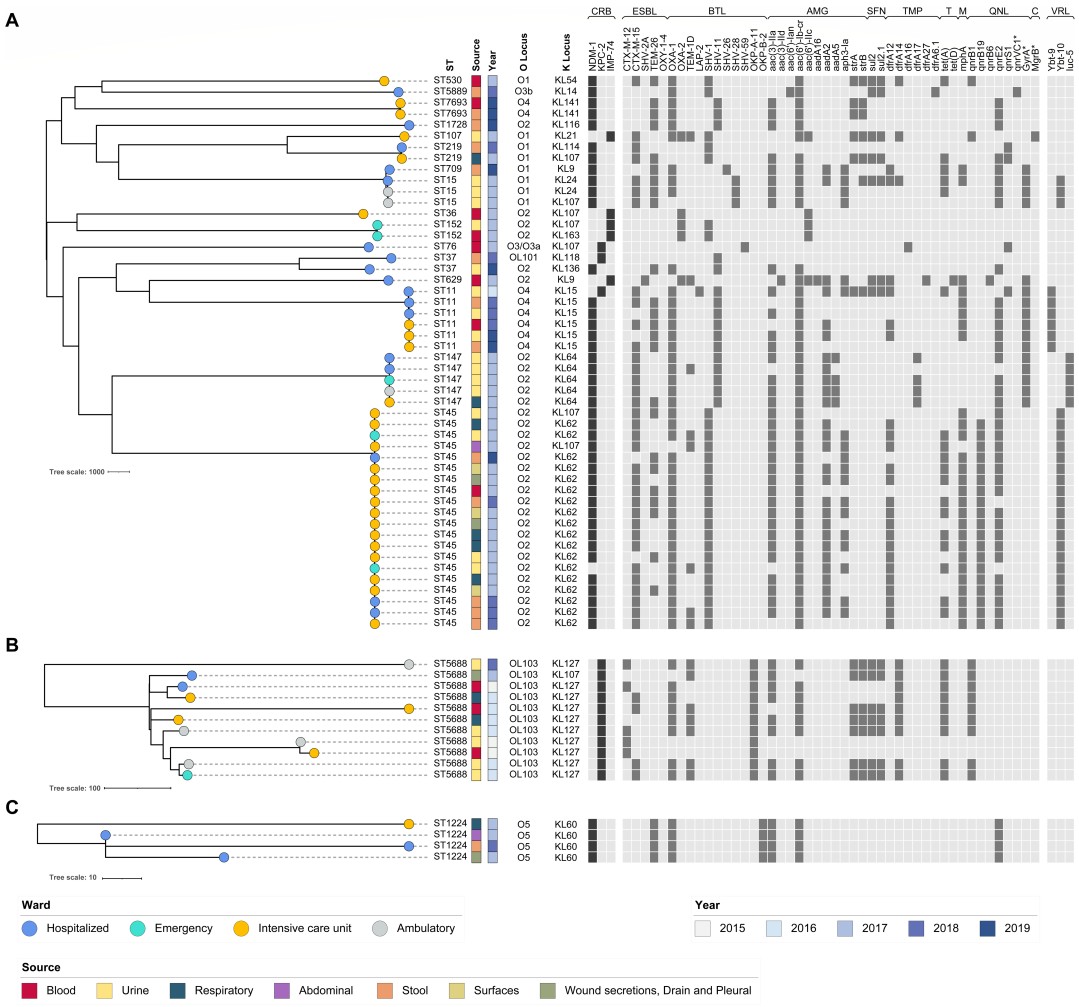

**FIG 3** Reference-based phylogeny, resistance, and virulence genes of isolates from this study: (A) *K. pneumoniae sensu stricto* (*n* = 50), (B) *K. quasipneumoniae* subsp. *quasipneumoniae* (*n* = 11), and (C) *K. quasipneumoniae* subsp. *similipneumoniae* (*n* = 4). ST: sequence type. CRB: carbapenems. ESBL: extended-spectrum beta-lactamase. BTL: beta-lactams. AMG: aminoglycosides. SFN: sulfonamides. TMP: trimethoprim. T: tetracyclines. MC: macrolides. QNL: quinolones. C: colistin. VRL: virulence genes.

isolates carrying the *bla*NDM-1 gene, included approximately one-fourth of the analyzed sequences, along with all three isolates recovered from hospital surfaces, and showed close genetic proximity between several isolates recovered from ICU and emergency department patients. There are two possible explanations for this finding: (i) it could be a result of a super-spreader event, supported by the finding that one isolate recovered from an ICU patient was the closest genetic neighbor to half of the sequences in the cluster or (ii) this ST45 clone has an inherited or acquired trait that increases its ability to spread in the hospital setting, possibly due to increased survival in hospital surfaces or increased capacity to colonize healthcare workers or patients. Supporting this last argument, ST45 has been previously implicated in CRKP hospital outbreaks in Europe, Asia, Africa, and the Middle East, particularly in neonatal units (41–45) and carrying a diversity of carbapenemase genes (*bla*KPC-3, *bla*OXA-48, *bla*GES-1/5, and *bla*NDM-1). Most recently, a Chilean hospital reported the emergence of ST45 carrying the *bla*NDM-1 gene during the COVID-19 pandemic (46). Although sequences from that study were unavailable during the preparation of this manuscript, future studies should assess the genetic relatedness between the Peruvian and Chilean ST45 strains.

Two prior studies analyzed CRKP outbreaks in two major Lima hospitals in 2016 and 2018 (15, 16). These studies identified *bla*NDM-1-carrying ST348 as the dominant lineage,

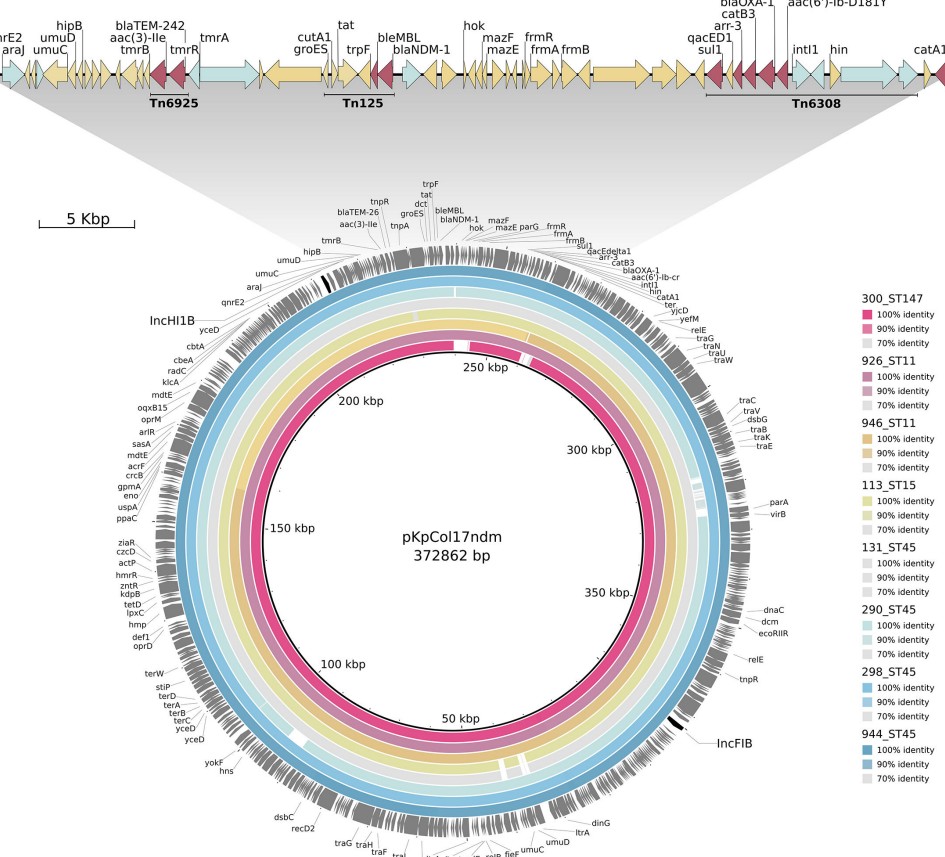

**FIG 4** Reconstruction of eight NDM-carrying plasmids from four *Klebsiella pneumoniae* sequence types. Top: detail of the genetic context surrounding the *bla*<sub>NDM-1</sub> gene. Red: antibiotic resistance genes; Cyan: mobile genetic elements; Yellow: genes with other predicted functions. Bottom: identity-based alignment with 100%, 90%, and 70% similarity thresholds against the previously published pKpCol17ndm conjugative plasmid reported in Peru (GenBank accession number CP072906).

followed by ST11 and ST147, and did not identify any ST45 isolates. In contrast, our study found ST45 as the predominant lineage, followed by ST11 and ST147, and did not detect ST348 isolates. Additionally, we found that *bla*<sub>NDM-1</sub> was carried on a large conjugative IncFIB-IncHI1B plasmid previously reported in other Peruvian hospitals (15, 37). The presence of this plasmid across multiple *Klebsiella* lineages and hospitals in Lima suggests that it confers an increased fitness and capacity for dissemination in the hospital environment. The co-carriage of genes encoding resistance to aminoglycosides and quaternary ammonium compounds supports this hypothesis, as these compounds are widely used for treating patients and for disinfection, respectively, in Peruvian hospitals (47).

Globally, CRKP epidemiology is marked by the regional predominance of high-risk clones, with ST258/512 being predominant in Europe and the USA and ST11 being most common in East Asia (12, 14). In Latin America, KPC-carrying ST11 and ST25 clones have expanded in hospitals across Brazil, Chile, and Colombia (10, 48, 49). Through a systematic search, we compiled a data set of 1,023 published CRKP genomes from South America and found a predominance of ST258/512 and ST11 isolates carrying KPC-like carbapenemases. Our results contribute to expanding the limited genomic data on CRKP available from Peru and confirm that multiple *bla*<sub>NDM-1</sub>-producing CRKP lineages have been established in Lima, including high-risk international lineages (ST11 and ST147) and emerging lineages (ST45 and ST348) that merit consideration from the global community.

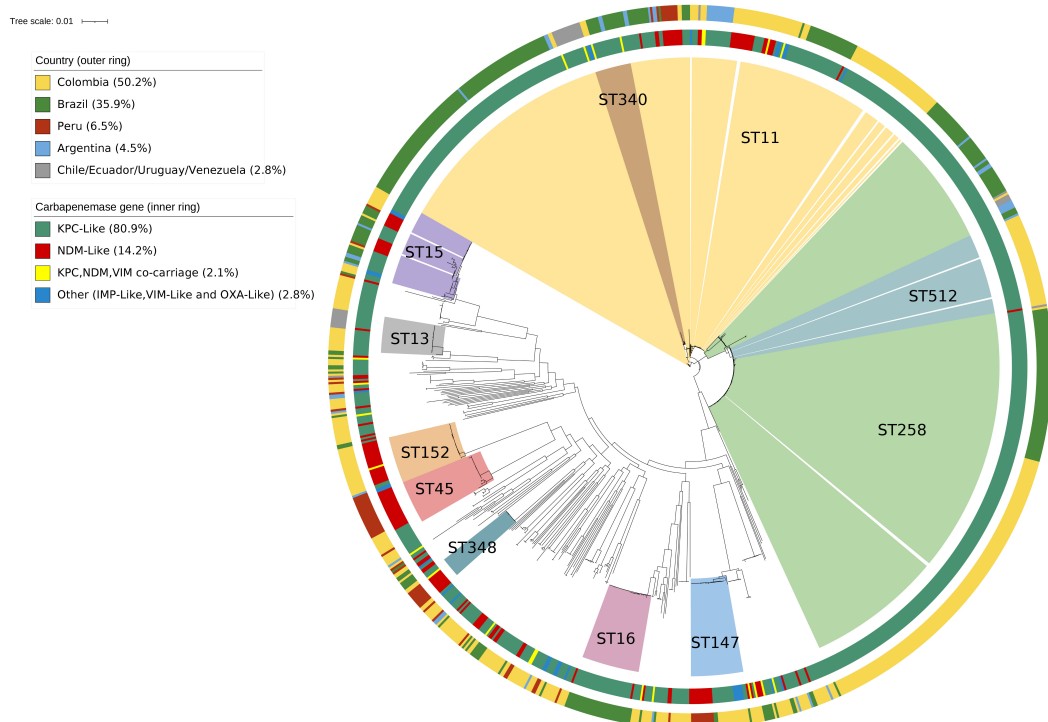

**FIG 5** Maximum-likelihood tree of 1,071 *K. pneumoniae sensu stricto* sequences from South America, including CRKP sequences from this study (*n* = 48) and other publicly available CRKP sequences from South America (2000–2021, *n* = 1023).

This study has several limitations. First, we only analyzed isolates that grew in the hospital microbiology laboratory as part of the routine patient care and intermittent surveillance, which may not fully capture the CRKP dynamics within the studied hospital. Second, CRKP isolates recovered during the pandemic and post-pandemic periods were unavailable for this study, limiting the results of this study to the pre-pandemic era. Future studies should expand to other healthcare facilities and include isolates collected during COVID-19 to assess potential shifts in CRKP transmission and resistance patterns. Finally, our plasmid analysis, conducted with long-read sequencing, was limited to only eight representative NDM-1 isolates. Although this offers a partial view of plasmid distribution, it provided sufficient information to confirm that a *bla*NDM-1-encoding plasmid has successfully established itself in Peru, transferring across various lineages and hospitals over different years.

This study contributes valuable genomic and epidemiological data on CRKP in Peru, highlighting a concerning shift toward *bla*NDM-carrying lineages and the emergence of high-risk clones such as ST45 and ST348. Our findings underscore the importance of regional surveillance and infection control strategies to monitor and contain the spread of these challenging pathogens. As the prevalence of *bla*NDM-producing CRKP continues to increase, coordinated efforts in antimicrobial stewardship, genomic surveillance, and infection control will be essential to mitigate the threat posed by carbapenem resistance in Peru and beyond.

## ACKNOWLEDGMENTS

We would like to thank the personnel of the Epidemiology Office and the Clinical Microbiology Laboratory of Hospital Cayetano Heredia for their assistance in data collection and Karen Ocampo for her assistance with the laboratory work of this study.

This work was supported by the Hospital Cayetano Heredia 2017 Research Grant; Prociencia (Grant No. 088-2018); Institut Mérieux under the Young Investigator Award to

F.K. and P.T.; and the Belgian Directorate of Development Cooperation and Humanitarian Aid (DGD) via the Institute of Tropical Medicine Antwerp. F. K. received a PhD Scholarship from the Belgian DGD through the Institute of Tropical Medicine Antwerp, as well as support by the Fogarty International Center of the NIH and the University of California Global Health Institute under Award Number D43TW009343. D.C. was supported by a training grant awarded to UPCH by the Fogarty International Center of the NIH under Award Number D43 TW007393.

## AUTHOR AFFILIATIONS

[1]Instituto de Medicina Tropical Alexander von Humboldt, Universidad Peruana Cayetano Heredia, Lima, Peru

[2]Facultad de Medicina Alberto Hurtado, Universidad Peruana Cayetano Heredia, Lima, Peru

[3]Department of Microbiology, Immunology and Transplantation, KU Leuven, Leuven, Belgium

[4]Laboratorio de Genómica Microbiana, Facultad de Ciencias e Ingeniería, Universidad Peruana Cayetano Heredia, Lima, Peru

[5]Department of Clinical Pathology, Hospital Nacional Cayetano Heredia, Lima, Peru

[6]Unit of Tropical Bacteriology, Department of Clinical Sciences, Institute of Tropical Medicine, Antwerp, Belgium

[7]Department of Infectious, Tropical, and Dermatological Diseases, Hospital Nacional Cayetano Heredia, Lima, Peru

[8]Wellcome Sanger Institute, Hinxton, United Kingdom

## AUTHOR ORCIDs

Fiorella Krapp  http://orcid.org/0000-0002-8404-2827
Guillermo Salvatierra  http://orcid.org/0000-0002-6887-2599
Pablo Tsukayama  http://orcid.org/0000-0002-1669-2553

## FUNDING

| Funder | Grant(s) | Author(s) |
|---|---|---|
| Hospital Cayetano Heredia | 2017 Research Grant | Fiorella Krapp |
| | | Catherine Amaro |
| | | Coralith Garcia |
| Prociencia | Grant No. 088-2018 | Pablo Tsukayama |
| Institut Mérieux (IM) | Young Investigator Award | Fiorella Krapp |
| | | Pablo Tsukayama |
| Belgian Directorate of Development Cooperation and Humanitarian Aid | | Fiorella Krapp |
| | | Coralith Garcia |
| HHS \| NIH \| Fogarty International Center (FIC) | D43TW00934 | Fiorella Krapp |
| HHS \| NIH \| Fogarty International Center (FIC) | D43TW007393 | Diego Cuicapuza |

## AUTHOR CONTRIBUTIONS

Fiorella Krapp, Conceptualization, Data curation, Funding acquisition, Methodology, Project administration, Supervision, Writing – original draft, Writing – review and editing | Diego Cuicapuza, Formal analysis, Software, Visualization, Writing – review and editing | Guillermo Salvatierra, Formal analysis, Visualization, Writing – review and editing | Jean P. Buteau, Data curation, Writing – review and editing | Catherine

Amaro, Conceptualization, Resources, Writing – review and editing | Lizeth Astocondor, Investigation, Methodology, Writing – review and editing | Noemí Hinostroza, Investigation, Methodology, Writing – review and editing | Jan Jacobs, Funding acquisition, Supervision, Writing – review and editing | Coralith García, Conceptualization, Funding acquisition, Methodology, Writing – review and editing | Pablo Tsukayama, Formal analysis, Funding acquisition, Methodology, Resources, Software, Supervision, Visualization, Writing – original draft

## DATA AVAILABILITY

Raw Illumina and Nanopore reads from carbapenem-resistant *Klebsiella pneumoniae* isolates generated in this study have been deposited under NCBI Bioproject PRJNA990326.

## ETHICS APPROVAL

This study was reviewed and approved by the Institutional Ethics Committee of Universidad Peruana Cayetano Heredia (N° 102526) and the Ethics Committee of Hospital Cayetano Heredia (N° 099-017).

## ADDITIONAL FILES

The following material is available online.

### Supplemental Material

**Supplemental material (Spectrum01825-24-s0001.docx).** Fig. S1 to S9; Table S1.

### Open Peer Review

**PEER REVIEW HISTORY (review-history.pdf).** An accounting of the reviewer comments and feedback.

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
