## [Reviewer comments · Microbiology Spectrum]

Microbiology Spectrum

Emerging carbapenem-resistant *Klebsiella pneumoniae* in a tertiary hospital in Lima, Peru

Fiorella Krapp, Diego Cuicapuza, Guillermo Salvatierra, Jean Buteau, Catherine Amaro, Lizeth Astocondor, Noemí Hinostrza, Jan Jacobs, Coralith Garcia, and Pablo Tsukayama

Corresponding Author(s): Fiorella Krapp, Universidad Peruana Cayetano Heredia Instituto de Medicina Tropical Alexander von Humboldt

Review Timeline:

Submission Date:	July 24, 2024
Editorial Decision:	September 1, 2024
Revision Received:	November 12, 2024
Accepted:	November 30, 2024

Editor: Silvia Cardona

Reviewer(s): Disclosure of reviewer identity is with reference to reviewer comments included in decision letter(s). The following individuals involved in review of your submission have agreed to reveal their identity: Zhen Shen (Reviewer #1)

Transaction Report:

DOI: <https://doi.org/10.1128/spectrum.01825-24>

Re: Spectrum01825-24 (Emerging carbapenem-resistant *Klebsiella pneumoniae* in a tertiary hospital in Lima, Peru)

Dear Dr. Fiorella Krapp:

Thank you for submitting your manuscript to Microbiology Spectrum. Your article has been reviewed by two experts in the field. Both reviewers found the findings interesting and have provide comments that, if addressed, will improve the quality of the work substantially.

Their recommendations are provided below.

Revision Guidelines

Sincerely,
Silvia Cardona
Editor
Microbiology Spectrum

Reviewer #1 (Comments for the Author):

This study evaluates the genomic diversity of CRKP isolates from a tertiary hospital in Lima, Peru. While CRKP lineages carrying blaKPC carbapenemase genes are prevalent across South America, this study interestingly identifies a predominance of four lineages carrying blaNDM carbapenemase genes in Peru. Although the study is compelling, I have a few concerns:

1. The CRKP isolates were collected between 2015 and 2019. Could more recent data, particularly from the COVID-19 pandemic period, be included in the study?
2. The study does not address the horizontal transfer of carbapenemase genes. A plasmid conjugation assay should be conducted to explore this aspect.
3. Lines 222-228: More detailed clinical data are needed to support the potential nosocomial transmission of CRKP. The genomic analysis suggests possible outbreaks of NDM-1-producing ST45 CRKP (Figure 3A).
4. Lines 244-245: The study does not clarify the genetic basis for colistin resistance in three CRKP isolates. Was colistin used to treat CRKP infections in these three patients?
5. Lines 252-253: For the ST147 CRKP isolates carrying iuc5, how does their virulence compare to ST147 CRKP isolates without iuc5? Are there any differences in symptoms and clinical outcomes among the patients infected with these different strains?

Reviewer #2 (Comments for the Author):

This study conducted an analysis of carbapenem-resistant *Klebsiella pneumoniae* (CRKP) strains isolated from a tertiary care hospital in Lima over the first five years (2015-2019). The research meticulously assessed the genetic diversity, resistance, and virulence profiles of these emergent strains. The collection of strains, informatics analysis, and biological experiments were rigorous and thorough, leaving no doubt about the validity of the results obtained.

RESPONSE TO REVIEWERS

Reviewer #1:

The CRKP isolates were collected between 2015 and 2019. Could more recent data, particularly from the COVID-19 pandemic period, be included in the study?

We agree with the Reviewer that it would be valuable to assess the distribution of *K. pneumoniae* isolates at this hospital during the COVID-19 pandemic and post-pandemic periods. Unfortunately, carbapenem-resistant *K. pneumoniae* isolates were not collected at our institution during these times. As such, we are unable to include this data in the current study. We have addressed this limitation in the Discussion section, emphasizing the importance of future studies that examine the post-pandemic distribution of *K. pneumoniae* in this and other healthcare facilities (L390–393).

The study does not address the horizontal transfer of carbapenemase genes. A plasmid conjugation assay should be conducted to explore this aspect.

We appreciate the Reviewer's suggestion to explore the horizontal transfer of carbapenemase genes. Due to technical limitations, we could not perform a plasmid conjugation assay in our lab. However, to provide evidence for the horizontal transfer of the *blaNDM-1* carbapenemase gene, we selected eight *blaNDM-1*-carrying isolates (one ST15, one ST147, two ST11, and four ST45) based on the *blaNDM-1* contig clusters identified from Illumina-based assemblies and conducted long-read nanopore sequencing to fully characterize their chromosomes and plasmids (L178–189). We have updated Figure 4 and revised the Results section (L305–310) to include these findings.

Our analysis presents three lines of evidence supporting the horizontal transfer of *blaNDM-1*:

1. All eight isolates contained a 372 kb plasmid harboring the *blaNDM-1* gene along with other resistance genes and multiple integrase, transposase, and mobilization-related elements. These plasmids were nearly identical across the isolates.
2. Using MOB-suite software, we determined that all full-length plasmid sequences belonged to the IncFIB-IncHI1B group and were predicted to be conjugative.
3. These plasmids showed high sequence similarity to a previously published 372 kb *blaNDM-1*-carrying plasmid from a *K. pneumoniae* ST147 isolate collected from another hospital in Lima in 2017. The authors of that study confirmed the transferability of the plasmid through a conjugation assay.

Lines 222-228: More detailed clinical data are needed to support the potential nosocomial transmission of CRKP. The genomic analysis suggests possible outbreaks of NDM-1-producing ST45 CRKP (Figure 3A).

We agree with the Reviewer that the genomic analysis suggests a potential outbreak of *blaNDM-1*-producing ST45 CRKP, supported by the very low SNP differences observed between several of these sequences. To complement the genomic data, we assessed two epidemiological criteria to support possible nosocomial transmission: (i) a temporal distance of less than three months

between isolate recovery and (ii) a shared ward of origin, defined as the hospital ward where the patient was hospitalized at the time of sample collection. We clarified these criteria in the Methods section (L137–140) and described in the Results section the proportion of ST45 sequences that met both criteria (L263–264).

However, the absence of electronic medical records in the hospital limits the clinical and epidemiological data that can be retrospectively accessed. Therefore, additional information that could further substantiate nosocomial transmission—such as overlap in hospital wards, surgical rooms, or procedure devices during patient hospitalization—was unavailable for assessment. This limitation has been addressed in the Discussion section (L347–349).

Lines 244-245: The study does not clarify the genetic basis for colistin resistance in three CRKP isolates. Was colistin used to treat CRKP infections in these three patients?

Our initial assessment of colistin resistance mechanisms included screening for the *mcr-1* gene and mutations in *mgrB* and *pmrB*. We have expanded this analysis to further investigate the genetic basis for colistin resistance in the four resistant isolates alongside a colistin-sensitive isolate and the reference *K. pneumoniae* strain NTUH_K2044. For these six genomes, we manually extracted and aligned coding sequences of the *pmrA*, *pmrB*, *phoP*, *phoQ*, *mgrB*, and *crrB* genes, which are known to carry mutations associated with colistin resistance. Our analysis identified resistance-associated mutations in *pmrB* (T157P, A246T, R256G) in three resistant isolates, while the fourth resistant isolate exhibited a V17E mutation in the *mgrB* gene and lacked the *crrB* gene entirely. We have added this in our Discussion section (L161-164) and Results section (L279-283).

Regarding treatment, only one of the four patients with colistin-resistant CRKP received colistin therapy, which was directed to treat a CRKP bloodstream infection; unfortunately, the patient died during treatment. The other three patients were not treated with colistin specifically for CRKP. Two were considered colonized by CRKP and received no targeted treatment; however, both had received colistin in the previous month to treat infections caused by different Gram-negative bacteria. The remaining patient did not receive colistin; he was given empiric treatment with meropenem and passed away before culture results were available.

Lines 252-253: For the ST147 CRKP isolates carrying *iuc5*, how does their virulence compare to ST147 CRKP isolates without *iuc5*? Are there any differences in symptoms and clinical outcomes among the patients infected with these different strains?

As indicated in Figure 3, all five ST147 CRKP isolates in this study carried the *iuc5* gene. Consequently, we cannot compare the symptoms and clinical outcomes between ST147 CRKP isolates with and without *iuc5* in our dataset, as no *iuc5*-negative ST147 isolates were available for analysis.

Reviewer #2:

This study conducted an analysis of carbapenem-resistant *Klebsiella pneumoniae* (CRKP) strains isolated from a tertiary care hospital in Lima over the first five years (2015-2019). The research meticulously assessed the genetic diversity, resistance, and virulence profiles of these emergent strains. The collection of strains, informatics analysis, and biological experiments were rigorous and thorough, leaving no doubt about the validity of the results obtained.

We thank the reviewer for the positive feedback.

Re: Spectrum01825-24R1 (Emerging carbapenem-resistant *Klebsiella pneumoniae* in a tertiary hospital in Lima, Peru)

Dear Dr. Fiorella Krapp:

I am pleased to let you know your manuscript has been accepted. Congratulations!

I am forwarding your manuscript to the ASM production staff for publication. Your paper will first be checked to ensure all elements meet the technical requirements. ASM staff will contact you if anything needs to be revised before copyediting and production can begin. Otherwise, you will be notified when your proofs are ready to be viewed.

PubMed Central: ASM deposits all Spectrum articles in PubMed Central and international PubMed Central-like repositories immediately after publication. Thus, your article automatically complies with the NIH access mandate. If your work was supported by a funding agency that has public access requirements like those of the NIH (e.g., the Wellcome Trust), you may post your article in a similar public access site, but we ask that you specify that the release date be no earlier than the date of publication on the Spectrum website.

Sincerely,
Silvia Cardona
Editor
Microbiology Spectrum